# Caveats to Exogenous Organic Delivery from Ablation, Dilution, and Thermal Degradation

**DOI:** 10.3390/life8020013

**Published:** 2018-05-12

**Authors:** Chris Mehta, Anthony Perez, Glenn Thompson, Matthew A. Pasek

**Affiliations:** School of Geosciences, University of South Florida, 4202 E Fowler Ave, Tampa, FL 33620, USA; camehta@mail.usf.edu (C.M.); aperez37@mail.usf.edu (A.P.); Thompsong@mail.usf.edu (G.T.)

**Keywords:** geology, cosmochemistry, astrobiology, ablation

## Abstract

A hypothesis in prebiotic chemistry argues that organics were delivered to the early Earth in abundance by meteoritic sources. This study tests that hypothesis by measuring how the transfer of organic matter to the surface of Earth is affected by energy-dissipation processes such as ablation and airbursts. Exogenous delivery has been relied upon as a source of primordial material, but it must stand to reason that other avenues (i.e., hydrothermal vents, electric discharge) played a bigger role in the formation of life as we know it on Earth if exogenous material was unable to deliver significant quantities of organics. For this study, we look at various properties of meteors such as initial velocity and mass of the object, and atmospheric composition to see how meteors with different initial velocities and masses ablate. We find that large meteors do not slow down fast enough and thus impact the surface, vaporizing their components; fast meteors with low masses are vaporized during entry; and meteors with low velocities and high initial masses reach the surface. For those objects that survive to reach the surface, about 60 to >99% of the mass is lost by ablation. Large meteors that fragment are also shown to spread out over increasingly larger areas with increasing mass, and small meteors (~1 mm) are subjected to intense thermal heating, potentially degrading intrinsic organics. These findings are generally true across most atmospheric compositions. These findings provide several caveats to extraterrestrial delivery models that—while a viable point source of organics—likely did not supply as much prebiotic material as an effective endogenous production route.

## 1. Introduction

The synthesis of prebiotic organic compounds is a key step in the origin of life, and sources of these materials are divided into endogenous and exogenous sources. Endogenous synthesis—occurring potentially on the surface of the earth—includes such historic experiments as the Miller spark discharge [1] as well as formose chemistry [2]. However, environmental factors appear to be an important restraint for endogenous prebiotic synthesis. For example, the lack of reduced gases such as H_2_ and CH_4_ would have greatly inhibited synthesis by spark discharge and the presence of atmospheric ammonia is predicted to have been lost quickly [3,4,5]. To this end, exogenous delivery of organics may have provided the key components of early prebiotic chemistry. Chyba and Sagan (1992) outline the three ways exogenous material on meteoric sources may have synthesized organic material on the early earth: extraterrestrial organics are synthesized prior to Earth entry through avenues such as radiation and delivered intact by meteorites; organic compound synthesis from ablation when entering Earth, and lastly by impact shock on the surface of Earth [6,7]. In this study, we examine how objects ablate through Earth’s atmosphere to provide better constraints on the delivery of exogenous organic matter.

Meteor ablation occurs when a meteor travels through the Earth’s atmosphere and the aerodynamic pressure of the surrounding atmosphere surpasses the material strength of the object. This generates heat from atmospheric entry and breaks up the object, causing it to lose mass. Although the net addition of mass to the earth remains the same during this process (e.g., a 1 kg meteoroid adds 1 kg to the total mass of the earth), the molecular and mineralogic constituents do not necessarily all survive (e.g., amino acids are thermally degraded and minerals are vaporized). This is because the temperature of ablation exceeds the melting and vaporization point of rock (1000–2000 K), typically sufficient to completely degrade organic compounds. If the objects are heated slowly, then some of these organic compounds may survive by sublimation [8], but more rapid heating may result in degradation instead. Therefore, to accurately measure the delivery of organic materials by meteoritic sources, it is imperative to consider how heating may influence organic speciation and delivery. Different meteoritic variables (composition, angle of entry, cross sectional area) and planetary variables (atmospheric density, scale height) are essential to understanding the ablation process.

Broadly, we distinguish three size groups of extraterrestrial material: large impactors, meteorites, and interplanetary dust particles/micrometeorites [9]. There is necessarily a continuum between these divisions, as differentiating between “large” micrometeorites and small meteorites, and between large meteorites and moderate-sized impactors is not straight-forward. However, as a rule, the Earth is subjected to a high rate of impact from smaller meteors; the more massive the object, the less frequently it impacts. Nevertheless, when these massive objects do impact the surface, they do so with almost all their initial mass still intact. Table 1 shows the rate of impact of meteors with varying initial masses, the amount of matter transported per Earth year, and the average time between impacts.

### 1.1. Prior Work

There are two groups of meteorites which are of interest to scientists studying transport of prebiotically-relevant matter through extraterrestrial sources: chondrites (especially carbonaceous chondrites) and iron meteorites (for phosphorus, see [9]). There are fifteen types of meteorites that fall within the chondrite group [10]. Of the fifteen, eight are classified as carbonaceous (of which four actually bear significant organics), three are classified as ordinary, and others fall within the subcategories of enstatite chondrites, R chondrites, and K chondrites [10]. Iron meteorites can be placed into fourteen major groups with about a remaining 110 objects ungrouped [11].

The organic compounds that have been found in carbonaceous chondrites consists of both a soluble fraction and an insoluble fraction, the latter is termed insoluble organic matter or IOM [12]. The soluble fraction includes a suite of amino acids, including biologic and non-protein amino acids [12,13], nucleobases [14], sugars and related compounds [15], simple hydrocarbons [16], and carboxylic acids [17]. The “gold standard” for most of these analyses is the CM chondrite Murchison, chosen because it was a relatively massive fall (~100 kg recovered), and soluble organics made up a significant fraction (~30%) of the total organic inventory of the meteorite. However, in recent times, significant work has been done on many of the other carbonaceous meteorites to elucidate variations in organic composition between varying groups and individual meteorites.

Within the organic-bearing carbonaceous chondritic meteorites there exists a broad continuum in many aspects of their organic geochemistry. Between the characteristics of the organic inventory (including distinct amino acid abundances), the fraction soluble organic inventory, organic chirality, and the oxidation state of organic material (e.g., [13,18,19]), meteorites show significant variation between individuals even of the same class. As such, there is significant variability in the presumed prebiotic relevance of various meteorite types: some may be extremely poor in relevant organics, others may be highly enriched in the key ingredients.

Intuitively, the delivery of organics to the surface of the early Earth requires that the organic compounds remain relatively unchanged from their synthesis in the interstellar medium or through aqueous alteration on their parent body to reaching the surface of the Earth. Ablation proves to be an obstacle in this area [20]. For an organic compound to reach the surface of Earth, it must survive thermal heating as it falls through the atmosphere. A meteor must also lose enough energy through both radiation and ablation during the descent to reach the surface of the Earth while remaining relativity intact [20]. Past studies have developed computational models that calculate the amount of mass lost by an object entering an atmosphere. For example, Bland and Artemieva (2006) modeled the ablation of meteors as they plummet through Earth’s atmosphere and as they impact the surface. This model is based on a series of equations that use physical characteristics of meteors that are crucial to solving for the ablation process. Further, the authors also model meteor fragmentation that occurs during the ablation process. These models, coupled with statistical methods, provide estimates of the impacting frequency of these objects to the surface of Earth. Bland and Artemieva conclude that chondrites that have an initial mass of greater than 50,000 kg impact the surface every 50 years.

Another significant study in ablation comes from Baldwin and Shaffer (1971) who developed a model to predict ablation of objects as they enter the Martian atmosphere. The authors do this to study how well the Martian atmosphere can protect the surface from impacts. The authors use ablation on Earth as a link to ablation on Mars; the only variables that needed to be changed in the models are the different atmospheric compositions/pressures and gravity of the two planets [21].

Hills and Goda (1993) discuss the importance of studying celestial objects traveling through our atmosphere and eventually impacting the surface of Earth. They focus on the consequences of the impact: the obvious of which is the loss of life and damage associated with the impact. The authors state that fragmentation is one of the main unknowns in this area of research and must be understood better by studying how objects entering the atmosphere fragment and where those fragmentations disperse. By studying objects of varying composition (such as iron, stone, comets, and disaggregated material), they conclude that for massive objects, low levels of ablation and fragmentation occurs [22].

Lastly, Chyba et al. (1993) write about the well-known impact over Tunguska, Siberia in 1908. In this study, they employ a computational model of different atmospheric conditions to better understand the impact of this large object, coupled with actual observed data from people living in the region at the time [23].

### 1.2. Meteor Energy Loss Due to Ablation

Depending on velocity, size, and composition a meteor entering the atmosphere could either explode (vaporize) before reaching the surface, or it could impact the surface [24]. Understanding energy properties of the ablating meteor is cardinal towards understanding the fate of the meteor as it travels through the atmosphere of Earth. Interestingly, the energy required to vaporize a meteor traveling at high speeds is much smaller than the initial kinetic energy of the meteor [25]. As a result, a meteor can completely vaporize if its kinetic energy is transformed into thermal energy. For instance, Zinn et al. (2004) proposed that the Leonid meteors vaporized before they notably decelerated from their entry velocity.

For those meteors that do not ablate completely, the major process that is involved in the dissipation of meteor energy in the atmosphere is the transformation of the kinetic energy to energy lost by ablation and heating of meteor mass. The energy of the ablated meteor is dissipated with celerity and a good portion of the mass of the meteor is in its tail, where the meteor breaks into finer particles [25]. The unablated meteor is intact and, if the size of the meteor is large enough (>1 mm), the interior temperature remains low.

However, a meteor that reaches the surface of the earth without decelerating sufficiently will vaporize on impact. If the velocity of the meteor on impact is sufficiently high, then, if the kinetic energy (KE = ½ mv^2^) exceeds the enthalpy of vaporization for rock (~10–15 MJ/kg), the meteor (and its organic constituents) should be expected to vaporize.

### 1.3. Thermal Diffusion and Meteor Ablation

Understanding the implications heating has in the ablation process of a meteor is still somewhat new and there is still much work in this area that needs to be done to have a better understanding on how the organics in a meteor may become altered by high temperatures during ablation. It is obvious that the exterior of a meteor is heated to the melting point of rock based on the presence of a glassy fusion crust, but the penetration of heat to the interior of a meteorite has conflicting reports of hot and cold from eye witnesses [26]. Many micrometeorites are also clearly melted (e.g., [27,28]), however, many more clearly survive entry unscathed [29].

During the ablation process, a meteor interacts with the atmosphere particles of Earth and if the conditions are right it will be shielded by vapors and particles of the meteor. Vinković [30] states that when the Knudsen number (K_n_, which is defined as the dimensionless ratio of the atmosphere mean free path and the size of the entering body) is less than 100, the particles and vapors surrounding the meteor act as a barrier which protects the meteor from atmospheric influenced ablation and, when this happens, one of the ways the meteor augments energy loss is though radiation.

### 1.4. Meteoroid Source Region and Distributions on Impact

Our knowledge of size distribution of asteroids and meteors—the parent source of meteorites—is still growing and there have been many attempts to understand the rate of Earth impacts for these objects [31,32,33,34]. In our Solar System, the area of that is most densely populated with asteroids is located 1.8 to 4.0 Astronomical Units (AU) from Earth [35]. Other areas with debris include the Kuiper Belt and the Oort Cloud, which is further towards the edges of our Solar System [35]. Lastly, there are objects close to Earth called Near-Earth Objects [35]. Per Britt et al. (2014) 1036 Ganymed is the largest Near-Earth Object and has a diameter of about 38.5 km. Smaller objects greatly outnumber larger objects [36], but the larger objects may have more mass than all the smaller objects combined.

In our study, we model meteors with initial masses ranging from 0.02 kg to 170,000 kg with varying initial velocities as they enter the atmosphere of Earth to determine how they ablate and if the organics in the meteor can survive the ablation process. The initial velocities are varied from 10,000 m/s to 42,000 m/s. The former corresponds to Near-Earth Objects that begin with a velocity equal to the escape velocity, or the velocity an object effectively has when captured by the gravitational pull of the earth. The higher velocities correspond to meteoroid source regions in the outer asteroid belt, which presumably is more likely to be carbonaceous.

If the meteor can successfully survive the atmospheric entry process, then the main mass along with pieces that were fragmented off it while making the transition from being a meteor to meteorite provide organics over a confined region. Therefore, to study how concentrated an area is with meteorites, we examine the relationship of strewn field areas and meteorite masses. If an area can be highly concentrated with fragments of meteorites, then the amino acids and other organics present in the object could in fact interact with the surrounding environment to aid in prebiotic chemical processes present on the early Earth.

## 2. Methods

We measured how the transfer of organic matter to the surface of the Earth is affected by ablation using a Riemann sum approximation. When a meteor ablates through Earth’s atmosphere the pieces are dispersed laterally and create an elliptical profile of fragments. Strewn fields are areas, often elliptical by nature, in which fragments of meteorites are found. Therefore, we also examine strewn fields and determine if there is a relationship between the total area of the field and the total recovered meteoric mass, and the implications for exogenous sources in prebiotic chemistry.

In addition, we consider how small meteors are affected by the diffusion of heat as the exterior of a meteor is ablated. We also take into account different variants of the prebiotic atmosphere in terms of molecular abundance of nitrogen and carbon dioxide. Though many researchers study these elements in terms of how they aided to life on Earth in terms of climate and ecosystems [37,38,39], we study it to understand how they affected the ablation process of meteors along with how the surviving pieces could perhaps concentrate an area with organics.

### 2.1. Ablation Model

The ablation model solves two equations describing velocity and change of mass (Equations (1) and (2) below). The atmospheric density is that of Earth and was set to 40.1 moles per meters cubed, and decreases with height according to the scale height (e.g., e-folding distance, or ~6–7 km). Bland and Artemieva (2006) describe a set of differential equations used to model ablation of meteors.
(1)$$\frac{dV}{dt}=-C_{d}\frac{\rho_{g}AV^{2}}{m}+g\;sin\left(\theta \right)$$
(2)$$\frac{dM}{dt}=-A\frac{min\left[\frac{c_{h}\rho_{g}V^{3}}{2},\sigma T^{4}\right]}{mQ}$$
where *V* is the initial velocity of the meteor, *m* is the mass of the meteor, *c_d_* is the Drag Transfer Coefficient, *c_h_* is the heat transfer coefficient, *ρ_g_* is the density of Earth’s atmosphere, *A* is the cross-sectional area of the meteor, *g* is gravity on Earth, *Q* is the heat of ablation, *σ* is the Stephan-Boltzmann constant, *T* is temperature, and *θ* is the angle the meteor enters Earth’s atmosphere. The initial distances for all meteors are set to 200 km and the angle of entry for all the simulations was set to 45 degrees. The distance (*D*) is calculated by the following calculation:(3)$$D=D_{N-1}-V_{N-1}\left(t_{N}-t_{N-1}\right)$$
where, *D_N_*_−1_ is the previous value of *D*, *V_N_*_−1_ is the velocity of the meteor at the previous distance, *t_N_* is the current time step and *t_N_*_−1_ is the previous time step. Initial mass of the meteors varied and all successive masses (M) were calculated by formula 2 where *M_N_*_−1_ is the calculated value of the mass preceding time steps.
(4)$$M=M_{N-1}-\frac{dM}{dt}\left(t_{N}-t_{N-1}\right)$$

For the purposes of this study and per Bland and Artemieva (2006), we consider only the flux of meteors that reach the surface that have masses greater than 0.02 kg. Therefore, only meteors that have a final mass of 0.02 kg were considered as part of the mass flux. Given these values, the model calculated the time it takes the object to impact Earth, the velocity along with the mass of the object at a given time, and the height-dependent atmospheric density encountered by the object as it travels through the atmosphere to the surface. The parameters used for these calculations are given as Table 2.

### 2.2. Strewn Field Analysis

A second consideration of meteoritic delivery is the actual concentration of meteoritic material over a geographic area. Since many meteorites fragment on entry, meteoritic masses are often spread over a region called a strewn field, confined to an ellipse where the masses are ultimately located. To determine if the surviving organic matter can be distributed in a large enough area to aid in prebiotic chemistry, strewn fields are also studied because of their role in the transport of organic matter through meteoritic sources. For the intents of this study, a strewn field is defined as an area, often elliptical, in which fragments of meteorites are found. MetBase, a meteorite information database software that catalogs the fall and the impact site of meteorites, is used to study the disbursement of fragments and organic matter from meteors.

### 2.3. Thermal Diffusion and Chemical Kinetics Model

One potential deleterious avenue that may affect semi-stable organic compounds within meteors is degradation through thermal diffusion of a meteor as it heats up and sheds mass, as it makes it transition from space to the surface of Earth [24,40]. We examine this by creating a computational model that calculates the heating profile occurring when a meteor is heated up by atmospheric entry. This is achieved by solving a differential equation (Equation (5)), which takes into account features such as thermal conductivity, temperature, radius of a spherical meteor, and time between the iterations of calculations performed. Equation (5) solves for the change in temperature with respect to radius where r is the radius in mm, *T* is the temperature in K, and a is the diffusion coefficient (set to 1 mm^2^/s. e.g., [41]).
(5)$$\frac{1}{r^{2}}\frac{\partial }{\partial r}\left(r^{2}\frac{\partial T}{\partial r}\right)=\frac{1}{\alpha }\frac{\partial T}{\partial t}$$

To this temperature-time (T-t) profile we add the kinetics of the decay of amino acid from Yaboklov et al. (2013) [42] wherein the thermal stability of valine and other amino acids were investigated to determine the rate of their decomposition as a function of time. We use valine as a specific proxy for other amino acids in carbonaceous meteorites, although Yaboklov et al. (2013) demonstrate little variation between different amino acids.

The decomposition rates were found to be 1st order with respect to amino acid quantity in the solid phase, and hence are likely related well to the phases found in chondrites. These data provide Arrhenius equations such as:(6)$$\ln k=1.78\times 10^{11}e^{\frac{-160}{RT}}$$
where R is 0.008314 kJ/mol K and T is the temperature in Kelvin. Using this data, we approximate the thermal stability (given as units of timescale to degrade 99% of the organics) of organics in meteors ranging in size from 0.1 mm to 10 mm heated for 1 s and with thermal diffusivity constants of 0.1 mm^2^/s to 10 mm^2^/s with an outside edge heated to 1000 K, an estimate of the melting point of chondritic material.

## 3. Results

On average, objects traveling at initial speeds of 10,000 m/s all reach the surface with ~50% of their mass so long as they are at least 0.05 kg before entering the atmosphere. Objects traveling at speeds of 14,000 m/s impact the surface with ~25% of their initial mass intact (Figure 1). Meteors with initial velocities of 18,000 m/s reach the surface with only 11–17% of their initial mass. If the initial velocity is 22,000 m/s, objects are subjected to more heating, thus more readily ablate and reach the surface with 3–5% of their initial mass. As the initial velocity increases, a meteor is subjected to even more ablation. For example, meteors with initial velocities between 26,000 to 38,000 m/s reach the surface with only 0.9 to 0.0001% of their starting mass. Figure 1 provides a summary of the different initial masses and velocities used along with calculated normalized final masses (mass at the surface divided by initial mass of meteor). What is seen is that for a meteor to have a final mass that is 0.02 kg or greater, it must have an initial mass of at least 0.05 kg and be entering the atmosphere of Earth at 10,000 m per second or less. As the initial velocity of a meteor increases, the final mass decreases (summarized as Table 3).

The initial mass is a greater control on final mass than the initial velocity overall: meteors of sufficiently large mass do not experience significant ablation and hence have not lost much mass when they reach the surface of the earth. This is mostly true independent of initial velocity, as meteors of sizes 10,000 to 100,000 kg and greater do not show much variation in mass on impacting the earth at any velocity. However, these objects are generally traveling at extreme speeds when they reach the earth’s surface and explode on impact. The initial velocity is an important factor for those meteors that do not explosively impact the surface. A meteorite’s final mass varies over several orders of magnitude depending on initial velocity for those objects that decelerate sufficiently to reach the surface of the earth going at terminal velocity or less.

We also find, in general, the effect of changing the atmosphere composition is that equivalently-sized meteorites ablate about the same in modern day atmosphere as they do in thicker atmospheres and thinner atmospheres (0.01–1000 kg initial mass meteoroid, velocity < 30,000 m/s, Figure 2 and Figure 3). However, a thick CO_2_ atmosphere allows for greater aerobraking of meteors that are large (up to 1,000,000 kg initial mass), whereas under typical atmospheric conditions meteors are not ablated if they have masses above about 50,000 kg, and reach the surface of the earth with most of their initial velocity. The thinner N_2_ atmosphere does not aerobrake meteors much above 10,000 kg in initial mass. In addition, meteors with high initial velocities are more likely to vaporize as they fall through the atmosphere under a thicker CO_2_ atmosphere, due to high frictional heating.

### 3.1. Strewn Field Mass-Area Relationship

Over one hundred strewn fields ranging from small to large (radial area of 1000 km^2^ and greater) were studied. We considered the amount of mass of meteorites in kilograms that were recovered in these fields. When we analyze the different strewn fields found on Earth and compare them the total recovered meteorite mass, there is a relationship between strewn field radial area and total mass recovered (Figure 4).

### 3.2. Radiation Model Results

The thermal profile of a 1 mm radius meteor with an exterior heated to 1000 K for one second is shown as Figure 5. It is apparent from this calculation that the meteor is completely heated to 1000 K to its core over this short timescale (we arrive at 1 s from the timescale of peak mass loss for most meteors). In contrast, the kinetic stability of amino acids is extremely short at 1000 K. At this temperature, 99% of amino acids degrade completely in about 10^−4^ s, implying little survival of amino acids during this heating event (Figure 6).

## 4. Discussion

We find meteorites to be small fragments of their initial mass for all initial velocities and mass ranges, losing a majority of their mass to high-temperature ablation. With the results, we propose several key caveats to exogenous delivery, and attempt to provide a numerical estimate of the amount of organics provided by meteorites, for comparison to other sources.

### 4.1. Caveats for Delivery

We find exogenous organic delivery needs to assumed cautiously (Figure 7). First, most of the mass of meteoroids is lost upon atmospheric entry. Organics within the ablated mass should be expected to be completely oxidized or polymerized to tar as the temperature exceeds the melting point of rock (> 1000 K, Figure 6). For large objects, the effects of ablation are not as prominent as they are for small to medium size objects. For small objects (having an initial mass of 1 kg and less), ablation is an evaporative process, completely disintegrating the object especially at high initial velocity. Intuitively, this makes sense; the smaller the meteor, the more it abates. Larger meteors have enough strength to withstand the ablation process and impact the Earth with significant mass. When vaporization of a meteor occurs, all the organics that were on the meteor can be assumed to be destroyed.

In contrast to the ablation and effective evaporation of small meteors going at high speeds, large meteors do not ablate significantly. Correspondingly, they also do not decelerate significantly. As a result, our second caveat is that these objects impact the earth’s surface at high velocity, often a velocity high enough to have more than the internal energy necessary to vaporize rock (≳ 10 MJ/kg, [43]). If a large object strikes the earth’s surface at greater than 4000 m/s, then it is highly likely that the object will completely vaporize on impact, and all minerals and organics contained within are reduced to gases. Both fast and large (massive) meteors would vaporize on impact.

The effect of various atmospheres on mass loss through ablation is not as huge as might be expected. A thin nitrogen atmosphere would have resulted in ablation similar to the modern atmosphere, except for less deceleration of large meteors. A thick CO_2_ atmosphere results in greater deceleration of larger meteors but somewhat higher total ablation of meteors as they travel through the atmosphere, especially at high initial velocity. The benefit of a thicker atmosphere on meteor survival is hence contingent on the mass-velocity distribution of the incoming bodies.

A potential route around the vaporization of meteors on impact is through fragmentation. Large meteors that fragment as they enter the atmosphere should ablate and decelerate if they do so 20,000–50,000 m above the surface. Fragmentation effectively allows ablation to act over a much larger surface area, enabling the deceleration and survival of the meteor on impact (with more organic survival). However our analysis of strewn fields suggests a third caveat: the size of strewn fields increases at a greater proportion than the increase in total mass. This implies that more massive meteors spread out over a larger area (thus providing less mass/area) than an equivalent smaller meteor. As a result, large objects are less effective as organic point sources than smaller meteors: a net dilution effect (e.g., [44]).

All the above seem to point to small meteors as being preferred sources of organics for exogenous delivery. Indeed, small meteors can lose energy through radiation in addition to ablation, thus they may reach the surface of the earth more readily. Our fourth caveat concerns small meteorites: we find the diffusion of heat to be rapid, and in the case of micrometeorites, it is sufficient to promote the complete degradation of organic molecules such as amino acids. A spherical meteor about 1 mm in radius will be completely heated to the exterior temperature (estimated here as 1000 K; see [45]), at which temperature it only takes 10^−4^ s to lose 99% of its amino acids. The interior of objects larger than 1 mm in radius may be more protected from this aggressive heating, and hence the organics in such objects should survive.

Although we present here a somewhat negative outlook on exogenous delivery, we acknowledge that there are several caveats to our own caveats. For one, a thick atmosphere results in more meteors reaching the surface of the earth intact. Secondly, the entry angle of meteors under consideration was assumed to be 45° (the average of 0 and 90°). Meteors with a more acute angle of entry suffer from less ablative effects as their vertical velocity is less. Finally, these models assume spherical objects. For some micrometeorites, this appears to be reasonable, but for most others, non-spherical shapes can alter the extent of ablation and of thermal heating. For instance, carbonaceous micrometeorites are rarely spherical [46], hence the thermal diffusion model presented here for those objects would be less applicable. However, many of the concerns raised here should be considered carefully in any putative prebiotic model that relies extensively on exogenous delivery for its key components.

### 4.2. Comparison to Natural Samples

The evidence of ablation on meteorites is recorded in their fusion crust (showing spatially-limited melting) and the shapes of recovered meteorites, which are occasionally oriented. It is difficult to provide physical comparison of meteors affected by ablation, but the process is well known in aerospace engineering, and has been simulated with artificial stones [47]. These simulations do demonstrate that mass is lost through ablation (77% lost), and that organics are significantly altered during descent through the atmosphere, transforming to graphite and losing ~66% of the total carbon. These results hint that organics are lost and ablated as meteors descend the atmosphere.

The effect of thermal processing on organics during ablation is recorded at least partially in spatially-resolved meteorite organic analysis, which demonstrate fusion crust are poor in thermally-labile organics [48,49], though analysis of fusion crust organics is not commonly done in meteoritics due to the possibility of contamination. Some micrometeorites show similar thermal processing, with the melted micrometeorites bearing effectively only graphite [46]. It should be noted, though, that not all micrometeorites melt on entry, and those that do not sometimes, but not always, bear extractable organic constituents [50].

### 4.3. Calculation of Exogenous Material Delivery to the Early Earth

The above caveats provide some constraints on the actual delivery of organic compounds to the surface of the earth. Using an assumed 10^22^ kg of meteoritic material accreting to the earth based on highly siderophilic elements (following [51]), we assume that the mass is distributed according to
Log N = −0.8 Log M + 17.1(7)
where N is the number of meteorites with mass greater than M, and the −0.8 comes from [36,52]. In such a case, the largest meteor to impact the earth would have been ~2.5 × 10^21^ kg, with masses decreasing by the log-log relationship from this largest mass as a starting point, and totaling 10^22^ kg. To this data, we apply the ablation survival calculations, assuming an average velocity of 18,000 m/s, and find that meteors of about 50,000 kg will survive without complete vaporization on landing (10% or 5000 kg reaches the surface of the earth).

We find a total mass of about 5 × 10^17^ kg delivered in this size range, and reaching the surface of the earth. To calculate the organic flux, this mass of meteorites is assumed to be 5% carbonaceous chondrites, with 2.2 wt.% carbon, of which 10% are soluble organic compounds [9], giving a total organic delivery of about 5 × 10^13^ kg, or about 10^14^ kg in the thick CO_2_ atmosphere. Spread across the surface of the earth this would correspond to about 10^5^ kg/km^2^ of total organics delivered over this bombardment period, or about 100 g/m^2^.

This high number would suggest that exogenous delivery may still be significant even with all the caveats we outlined above, but it must be noted that this is the total flux, integrated over the several hundred million years assumed for delivery by [51]. This number is also the total soluble organics, and not of individual molecular functional groups (e.g., not just amino acids, but including carboxylic acids, alcohols, hydrocarbons, and others). In contrast, over the same amount of time electric discharge is expected to form ~10^15.5^–10^17.5^ kg of organics [7]. Nonetheless, these data do hint that exogenous delivery, even with several cautionary caveats, may act as a significant net organic source.

## 5. Conclusions

We have outlined several cautionary points to assuming meteorites were principal sources of organics to the prebiotic earth. The atmosphere acts as an effective filter for most meteors, vaporizing the fastest ones and many of the small meteors. Large meteors reach the surface but may do so explosively (vaporizing organics), whereas fragmentation of large meteors scatters material over an ever-increasing region. Small meteors may lose energy by other processes, but even with a short heating event from deceleration, effectively all organics can assume to be significantly altered from their initial, prebiotically-relevant form. With these filters, we estimate other endogenous sources may have been more important, though meteorites could have certainly been locally relevant, and may have supplemented key nutrients on the early earth.

## Figures and Tables

**Figure 1 life-08-00013-f001:**
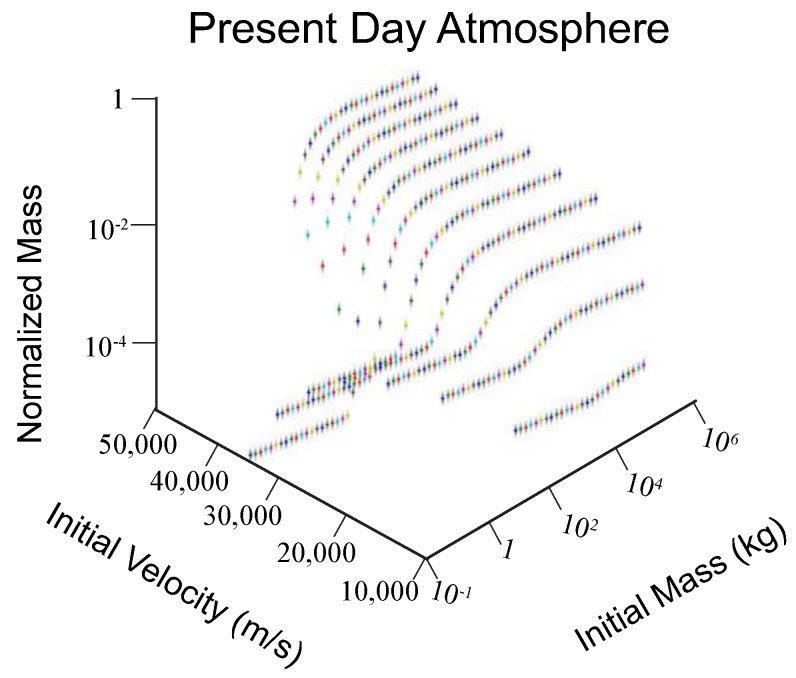
The final mass of a meteorite vs. its initial mass and velocity, under present atmospheric conditions. Normalized mass is the final mass of the object divided by its initial mass.

**Figure 2 life-08-00013-f002:**
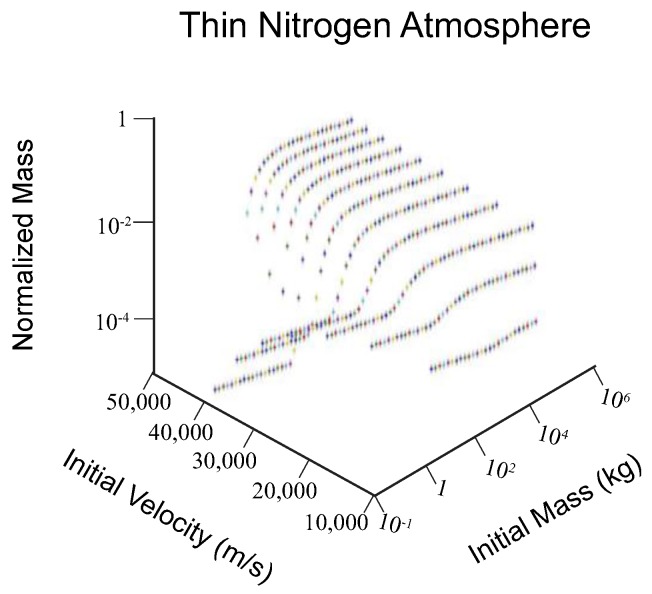
As per Figure 1, for a thin N_2_ atmosphere (0.5 atm at the surface).

**Figure 3 life-08-00013-f003:**
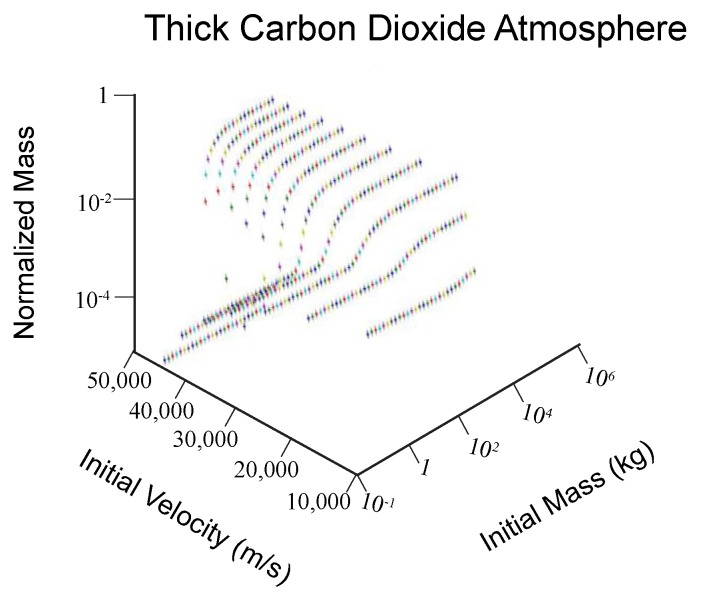
As per Figure 1, with a CO_2_ atmosphere (3 atm at the surface).

**Figure 4 life-08-00013-f004:**
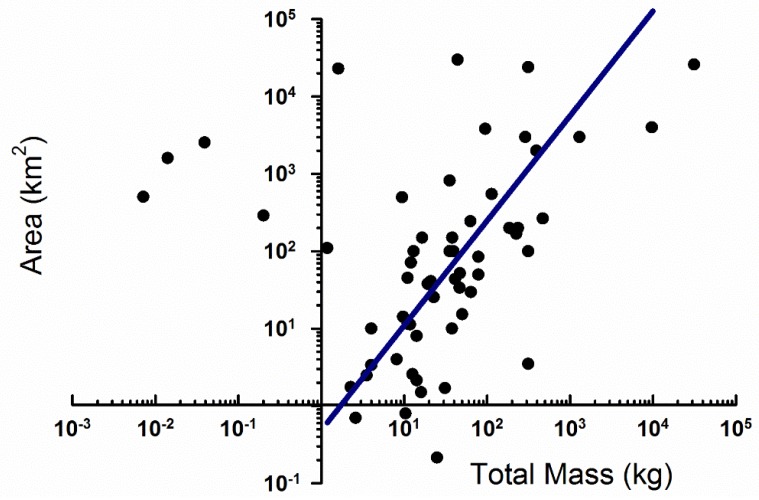
Relationship between the total mass of a single meteorite recovered, and the area over which all the fragments were recovered (the strewn field). The regression line has a slope of log A = 1.25 log M − 0.23 (R^2^ = 0.47) where A is the radial area and M is the mass. This is not to say that the mass is the direct control on strewn field area, as other factors (angle of entry, meteoroid structural integrity prior to entry) likely play as large of roles in strewn field sizes. Some of this information is not known for these strewn fields as many are meteorite finds. However, that there is a relationship between mass and area is suggestive that the mass controls a portion of the area over which a meteorite is distributed.

**Figure 5 life-08-00013-f005:**
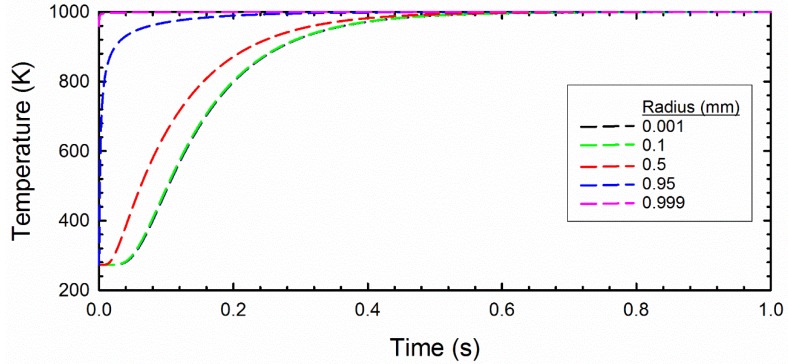
T-t-r (Temperature, time and radius) profile predicted for meteors in contact with a 1000 K surface. This model assumes a total meteor radius of 1 mm, and the dashed lines provide the temperature from the center (0 mm) to the exterior (1 mm) as a function of time.

**Figure 6 life-08-00013-f006:**
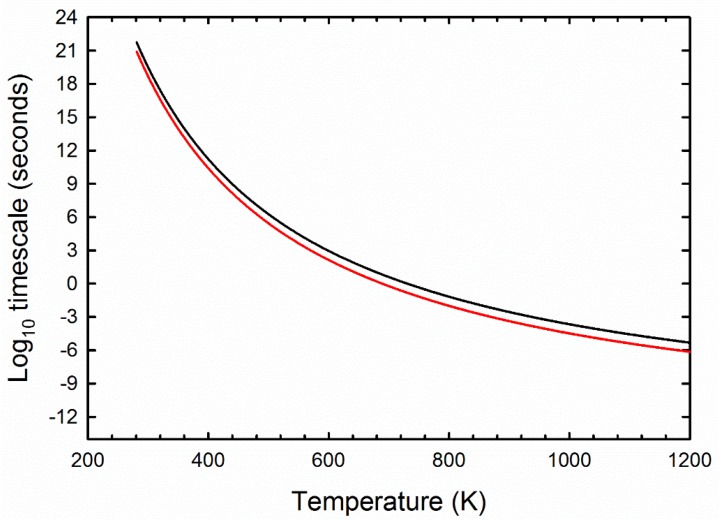
The timescale it takes to degrade valine as a function of temperature. The black line is the time required to degrade 50% of the material (a chemical half life), and the red line corresponds to the time required to degrade 99% of the valine.

**Figure 7 life-08-00013-f007:**
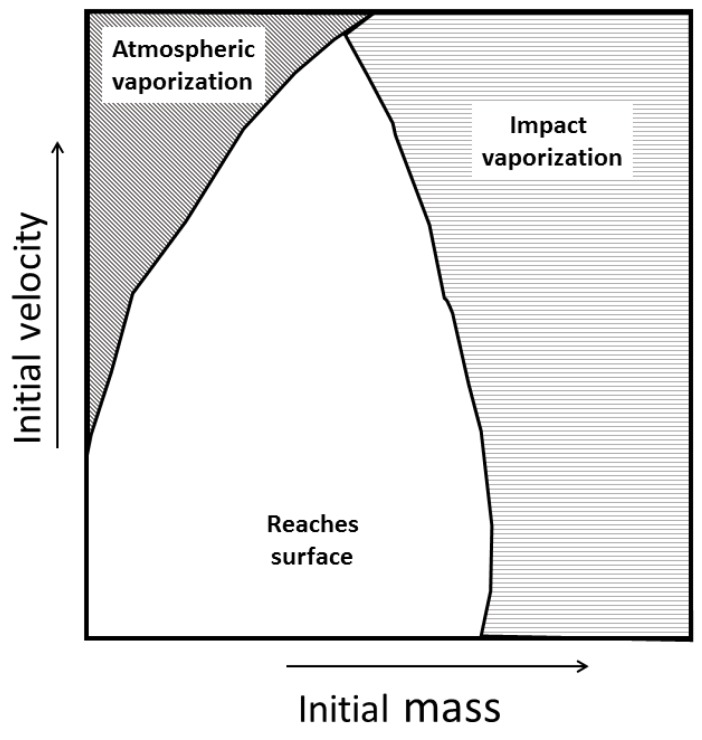
Schematic showing that higher velocity objects may vaporize as they travel through the atmosphere, and larger objects vaporize when they impact the surface.

**Table 1 life-08-00013-t001:** Rate of meteor impact on earth. Table adapted from Bland and Artemieva (2006) that shows the delivery by mass to the earth per year of meteors with specific initial masses, and the interval between falls of a given mass. For example, meteorites with an initial mass of 1 kg fall roughly four times every hour (0.00003 years), delivering a net 37,000 kg/year.

Mass (kg)	kg/year	Intervals between Years
0.10	110,000.00	0.00001
0.50	52,000.00	0.00002
1.00	37,000.00	0.00003
5.00	12,000.00	0.00008
100.00	800.00	0.00130
500.00	170.00	0.01
1000.00	91.00	0.01
5000.00	21.00	0.05
10,000.00	11.00	0.09
50,000.00	2.40	0.42
100,000.00	1.30	0.77
500,000.00	0.29	3.46
1,000,000.00	0.15	6.58
5,000,000.00	0.02	41.32

**Table 2 life-08-00013-t002:** Parameters used in ablation model.

Parameter	Symbol	Value
Atmosphere Surface Density	ρ_atm,Present day_	40.1 moles/m^3^
Scale Height Present Day	H	7 km
Thin N_2_ Atmosphere Surface Density	ρ_atm,N2_	20 moles/m^3^
Scale Height	h_N2_	7.5 km
Thick CO_2_ Atmosphere Surface Density	ρ_atm,C02_	120 moles/m^3^
Scale Height	h_CO2_	5 km
Starting Altitude	Z	200,000 km
Drag Coefficient	c_d_	0.47
Angle of Entry	θ	45°
Density	ρ	3.5 g/cc
Ablation Coefficient	c_a_	0.014
C_h_/2Q	N/A	6.25 × 10^−9^ s^2^/m^2^

**Table 3 life-08-00013-t003:** Ablation under current atmospheric conditions. Values are the minimum initial mass for a meteoroid before atmospheric entry for impact to occur for Earth’s modern day atmosphere.

Initial Velocity (m/s)	Minimum Initial Mass (kg)
10,000	0.05
14,000	0.1
18,000	0.2
22,000	0.72
26,000	2.64
30,000	13.5
34,000	132
38,000	13,000

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
