# Peer review of "Caveats to Exogenous Organic Delivery from Ablation, Dilution, and Thermal Degradation"

_life, 2018, doi:10.3390/life8020013_

Round 1
Reviewer 1 Report
This is an excellent and timely paper. Given that there have been several recent papers published that use extraterrestrial delivery of organic compounds as the base for their study, this will add a new important limitation in future studies. The only suggestion I have is that a previous study (below) showed that some amino acids in small meteorite fragments may sublime and escape heat destruction.
Glavin, Daniel P., and Jeffrey L. Bada. "Survival of amino acids in micrometeorites during atmospheric entry." Astrobiology 1.3 (2001): 259-269.
Author Response
Thank you. We have added a reference to this paper in the introduction, as it provides a good work-around for the problems we propose for organic delivery.
Reviewer 2 Report
Dear authors,
I have read with a great attention your manuscript entitled “Caveats to Exogenous Organic Delivery from Ablation, Dilution, and Thermal Degradation” and, although I found the subject interesting, I found that your manuscript must be improved prior any publication.
It is not very well organized. In particular, the name of the different parts is not necessarily well chosen and the subsections 1.1, 3.1 and 4.1 appear to far from the title of their parent section. Moreover, the part 4.1 is not relevant since there is no part 4.2. Your manuscript has no conclusion.
Some sentences and references are not necessary to the study. They are confusing and impair the understanding of the study. On the contrary, some references are missing. In particular there is no reference about the organic molecules actually found in meteorites such as Murchison for example.
The figures are not well presented and difficult to understand. The tables 1 and 2 could be improved. The equations are not all numbered and badly displayed.
The subject of the study is interesting but the part dedicated to the survival of organics is too small. The result of the modelling must at least be compared to natural samples.
As you can read, I am a little bit reserved about your manuscript for which I suggest major revision prior publication.
Best regards
Author Response
Our responses are preceded by ">>"
It is not very well organized. In particular, the name of the different parts is not necessarily well chosen and the subsections 1.1, 3.1 and 4.1 appear to far from the title of their parent section.
>> These sections have been modified and edited for focus on organics, as requested. We have renamed the subsections to better match their topic.
Moreover, the part 4.1 is not relevant since there is no part 4.2.
>> We have broken out our discussion into a second section with “caveats” as part 4.1, and 4.2 is a comparison to natural samples, and 4.3 as the mass calculation
Your manuscript has no conclusion.
>>Now added.
Some sentences and references are not necessary to the study. They are confusing and impair the understanding of the study.
>> There is some material in section 1.1 that we now term “prior work” that give the information on how ablation works. We have trimmed other references as well.
On the contrary, some references are missing. In particular there is no reference about the organic molecules actually found in meteorites such as Murchison for example.
>>We have added two paragraphs specifically detailing the organic inventory of carbonaceous meteorites, and the molecules found therein.
The figures are not well presented and difficult to understand.
>> The figures have been redone to improve their clarity and the captions have been modified with better descriptions.
The tables 1 and 2 could be improved.
>>We have added descriptive text to table 1's caption, and a second column to table 2 that provides the symbols correlating to the descriptive text and assumed values.
The equations are not all numbered and badly displayed.
>>We’ve gone through and numbered the equations that escaped numbering in the prior version. However, we do not see the bad display of these equations. Is it possible that these were modified in conversion to pdf? We will try to ensure that these equations are clear in the next uploaded version.
The subject of the study is interesting but the part dedicated to the survival of organics is too small. The result of the modelling must at least be compared to natural samples.
>> We now compare the results to natural samples in section 4.2, and increase our focus on organics throughout the paper.
Round 2
Reviewer 2 Report
Dear authors,
I have read the revised version of your manuscript entitled “Caveats to Exogenous Organic Delivery from Ablation, Dilution, and Thermal Degradation” and, although I found the subject interesting, I found that your manuscript must still be improved prior any publication.
In particular, you wrote in your answer that the figures have been redone to improve their clarity, however, I still find them relatively bad. First, they are not standardized (i.e. font, size… are different).
Most importantly, Figures 1, 2 and 3 appear in contradiction with the text and with the table 3; the curves displayed show that the loss of mass is all the more important than the velocity is small. These figures are very difficult to read. As a comment, 10^0 can be replace by 1.
Figure 4 is not very nice too. The linear regression must be removed (R²= 0.47 means that it is not linear!).
Figure 6 is just a copy and paste from Excel.
Even if it is obvious, add arrows to display the increasing direction of mass and velocity on Figure 7.
For the equations, they are worthily displayed than in the previous version. Like for the figures, the font size is not the same everywhere. This is particularly obvious on page 2 between the equations 5 and 6. Most of the symbols are not appearing in the text and in the Table 2.
Finally, if you increased the scientific part which is now OK, the quality of presentation remains too low and must be improved prior publication of your manuscript.
Best regards
Author Response
Dear authors,
I have read the revised version of your manuscript entitled “Caveats to Exogenous Organic Delivery from Ablation, Dilution, and Thermal Degradation” and, although I found the subject interesting, I found that your manuscript must still be improved prior any publication.
>> We thank the reviewer for the thorough review as it continues to refine the manuscript to make it more acceptable for this special edition. We have spent most of this review iteration modifying the figures. As a result, our figures 4-7 are all standardized and the axes of figures 1-3 match those of 4-7 with respect to line thickness and font. Please note that our access to the presentation of figures 1-3 is limited because of issues with the plotting program. Differences in size of the figures should be modifiable by the editorial office as these have been submitted as pdfs or tifs, and can be stretched within/external to the document.
In particular, you wrote in your answer that the figures have been redone to improve their clarity, however, I still find them relatively bad. First, they are not standardized (i.e. font, size… are different).
>> We have endeavored to standardize all the plots, but due to difference in data presentation (figures 1-3 are 3D representations of the data, whereas figures 4-7 are 2D), and differences in data presentation (fig 1-3 were generated by MatLab, the others by sigmaplot and power point [the schematic]) there’s some difficulty in presenting these equivalently.
Most importantly, Figures 1, 2 and 3 appear in contradiction with the text and with the table 3; the curves displayed show that the loss of mass is all the more important than the velocity is small. These figures are very difficult to read. As a comment, 10^0 can be replace by 1.
>> Just to make sure we’ve interpreted your critique correctly: do you mean that initial mass is a bigger influence on final mass than initial velocity? If so, we agree: the initial mass provides the strongest influence on final mass. We’ve added the following paragraph to the text in the results:
The initial mass is a greater control on final mass than the initial velocity overall: meteors of sufficiently large mass do not experience significant ablation and hence have not lost much mass when they reach the surface of the earth. This is mostly true independent of initial velocity, as meteors of sizes 10,000 to 100,000 kg and greater do not show much variation in mass on impacting the earth at any velocity. However, these objects are generally traveling at extreme speeds when they reach the earth’s surface and explode on impact. The initial velocity is an important factor for those meteors that do not explosively impact the surface. A meteorite’s final mass varies over several orders of magnitude depending on initial velocity for those objects that decelerate sufficiently to reach the surface of the earth going at terminal velocity or less.
>> We’ve adjusted these figures to match scales and font sizes.
>> We’ve replaced 10^0 with 1.
Figure 4 is not very nice too. The linear regression must be removed (R²= 0.47 means that it is not linear!).
>> An R2 of 0.47 is not great, but there are multiple factors that would influence strewn field sizes, including angle of entry, reporting accuracy on the strewn field size, and structural characteristics of meteoroid prior to entry (e.g., if it was already fragmented vs. if it was intact). Information on many of these controlling factors has been lost since collection, as several strewn fields are of meteorite finds and were not observed to fall. Hence that there is even a relationship between the two (R2 of 0.47 means that the strewn field size is roughly half attributable to its mass), we argue that this would merit inclusion in the data. If desired, we could remove the line if it distracts too much from the graph.
>>We have added the following:
>>This is not to say that the mass is the direct control on strewn field area, as other factors (angle of entry, meteoroid structural integrity prior to entry) likely play as large of roles in strewn field sizes. Some of this information is not known for these strewn fields as many are meteorite finds. However, that there is a relationship between mass and area is suggestive that the mass controls a portion of the area over which a meteorite is distributed.
Figure 6 is just a copy and paste from Excel.
>> Changed to match the standardization of other graphs. We’ve also altered this to provide a half-life as well as 99% degradation timescale.
Even if it is obvious, add arrows to display the increasing direction of mass and velocity on Figure 7.
>>Done
For the equations, they are worthily displayed than in the previous version. Like for the figures, the font size is not the same everywhere. This is particularly obvious on page 2 between the equations 5 and 6. Most of the symbols are not appearing in the text and in the Table 2.
>>This looks to have been fixed by the editorial staff.
Finally, if you increased the scientific part which is now OK, the quality of presentation remains too low and must be improved prior publication of your manuscript.
>>Thank you again!
Best regards